# Efficiently Anti-Obesity Effects of Unsaturated Alginate Oligosaccharides (UAOS) in High-Fat Diet (HFD)-Fed Mice

**DOI:** 10.3390/md17090540

**Published:** 2019-09-17

**Authors:** Shangyong Li, Ningning He, Linna Wang

**Affiliations:** 1Department of Pharmacology, College of basic Medicine, Qingdao University, Qingdao 266071, China; lshywln@163.com; 2Yellow Sea Fisheries Research Institute, Chinese Academy of Fishery Sciences, Key Laboratory for Sustainable Development of Marine Fisheries, Ministry of Agriculture, Qingdao 266071, China

**Keywords:** unsaturated alginate oligosaccharides, hepatoprotective effect, anti-oxidant activity, anti-obesity activity

## Abstract

Obesity and its related complications have become one of the leading problems affecting human health. However, current anti-obesity treatments are limited by high cost and numerous adverse effects. In this study, we investigated the use of a non-toxic green food additive, known as unsaturated alginate oligosaccharides (UAOS) from the enzymatic degradation of Laminaria japonicais, which showed effective anti-obesity effects in a high-fat diet (HFD) mouse model. Compared with acid hydrolyzed saturated alginate oligosaccharides (SAOS), UAOS significantly reduced body weight, serum lipid, including triacylglycerol (TG), total cholesterol (TC) and free fatty acids (FFA), liver weight, liver TG and TC, serum alanine aminotransferase (ALT), and aspartate aminotransferase (AST) levels, adipose mass, reactive oxygen species (ROS) formation, and accumulation induced in HFD mice. Moreover, the structural differences in β-d-mannuronate (M) and its C5 epimer α-l-guluronate (G) did not cause significant functional differences. Meanwhile, UAOS significantly increased both AMP-activated protein kinase α (AMPKα) and acetyl-CoA carboxylase (ACC) phosphorylation in adipocytes, which indicated that UAOS had an anti-obesity effect mainly through AMPK signaling. Our results indicate that UAOS has the potential for further development as an adjuvant treatment for many metabolic diseases such as fatty liver, hypertriglyceridemia, and possibly diabetes.

## 1. Introduction

Obesity is a genetic and health problem, as well as a social problem that seriously threatens human health. According to data from the World Health Organization (WHO) in 2016, 25% of adults worldwide were overweight [1,2]. Overweight and obesity can cause complications such as type 2 diabetes, high blood pressure, cancer, and non-alcoholic fatty liver disease (NAFLD) [3,4]. However, there are limited anti-obesity drugs on the market due to their unexpected side effects, such as increasing the risk of heart attacks and psychiatric side effects [5,6]. Currently, orlistat is the only over-the-counter drug approved by the Food and Drug Administration (FDA) to help with obesity. However, it also causes a series of adverse reactions in certain people, including gastrointestinal trauma and greasy stools [7]. Therefore, it is necessary to develop alternative anti-obesity agents, including safe and effective functional foods for adjuvant therapy.

Brown seaweed, especially *Laminaria japonica* (LJ), is one of the most commonly eaten seaweeds in the world, especially in Scandinavia and East Asia [8]. As a functional food, LJ and its extracts have been shown to have anti-diabetic, anti-inflammatory, anti-oxidant, and anti-cancer effects in several studies [9,10]. Alginate is the most abundant carbohydrate in brown algae and approximately 30,000 tons of alginate is produced annually worldwide [11,12]. Due to its good safety profile and effectiveness, alginate has been widely used in the the food and pharmaceutical industries. In nature, alginate is a linear hetero-polyuronic acid polymer, composed of β-d-mannuronate (M) and its C5 epimer α-l-guluronate (G) [12,13]. Recently, alginate oligosaccharides (AOS) have gained greater attention due to their important biological functions, such as their ability to improve the growth of bifidobacteria, and improve cytokine-inducing activity in mononuclear cells, antioxidant activity, and plant root growth-promoting activity [14,15,16,17,18]. Currently, there are two different degradation protocols to produce alginate. The traditional acid hydrolysis method yields saturated alginate oligosaccharides (SAOS) as the main products, while enzymatic hydrolysis using alginate lyase produces unsaturated alginate oligosaccharides (UAOS) as the reaction products [19].

Compared with SAOS, the non-reducing ends of UAOS provide double bonds between C4 and C5, which gives them greater anti-oxidant activity than ascorbic acid in the lipid oxidation treatment [19]. Obesity-induced accumulation of liver lipids and reactive oxygen species (ROS) is a key factor in the prevalence of metabolic and cardiovascular diseases [20]. Excessive liver ROS production and decreased cellular antioxidant activity lead to oxidative stress and liver oxidative damage [21]. Therefore, antioxidants can effectively improve lipid and ROS metabolism, which reduces liver steatosis and oxidative damage. However, the in vivo antioxidant, with hepatoprotective and anti-obesity activities of UAOS, have not yet been reported. Obesity is a disorder that is related to energy imbalance. AMP-activated protein kinase (AMPK) is a crucial cellular energy sensor. Once AMPK is activated, it triggers catalytic processes to generate ATP while inhibiting anabolic processes that consume ATP in an attempt to restore cellular energy homeostasis. AMPK is considered a potential target for treating metabolic disorders [22]. It has previously been reported that anti-obesity agents activated the AMPK signaling pathway, which is related to the suppression of adipogenic differentiation and lipogenesis AMPK, such as Rg1 [23], dioxinodehydroeckol [24], and ethanol extracts of Aster yomena [25]. Increasing evidence has demonstrated that AMPK can inhibit adipogenesis and suppress the expression of SREBP-1c, PPARγ, and FAS in adipocytes [26,27]. Therefore, in this study, we will study whether AMPK is involved in the anti-obesity mechanism of UAOS.

In this study, we investigated that UAOS showed effective anti-obesity effects, which significantly reduced body weight, serum lipid, including triacylglycerol (TG), total cholesterol (TC), and free fatty acids (FFA), liver weight, liver TG, and TC, serum alanine aminotransferase (ALT) and aspartate aminotransferase (AST) levels, adipose mass, reactive oxygen species (ROS) formation, and accumulation induced in HFD mice. This study also showed that UAOS played an anti-obesity effect mainly through an AMP-activated protein kinase α (AMPKα) signaling pathway. UAOS may be a promising candidate for the treatment of metabolic diseases such as fatty liver, hypertriglyceridemia, and possibly type 2 diabetes.

## 2. Materials and Methods

### 2.1. Materials and Supplies

High viscosity sodium alginate (20–50 kDa, 100–260 polymerized monosaccharides, M/G ratio: 1.66) was prepared. Standard alginate disaccharide and trisaccharide were purchased from Qingdao BZ Oligo Biotech Co., Ltd (Qingdao, China). Chitosan, with a ≥95% degree of deacetylation (DDA), were purchased from Aladdin, China. Antibodies against p-AMPKα (Thr172), AMPKα, p-ACC (Ser79), ACC, and GAPDH were purchased from Cell Signaling Technology (Beverly, MA, USA). Orlistat was purchased from Aladdin (Aladdin, Shanghai, China). A standard diet (STD) feed and a high-fat diet (HFD) feed were purchased from the Darenfucheng Animal Husbandry Company (Qingdao, China). The nutritional composition of a standard diet (STD) and a high-fat diet (HFD) were shown in Appendix A. All chemical reagents used in this study were of an analytical grade.

### 2.2. Preparation and Determination of Oligosaccharides

Polyguluronate block (Poly G) and polymannuronate (Poly M) (DP = 20–24) were prepared from sodium alginate, according to the method of Haug et al. [28]. Two types of enzymatic unsaturated alginate oligosaccharides (UAOS) (unsaturated mannuronate oligosaccharides (UMOS) and unsaturated guluronate oligosaccharides (UGOS)) were prepared by using an alginate lyase purified in our laboratory [29]. The reaction mixture contained ~20,000 U (50 mL) of bifunctional alginate lyase Aly08 and 1 L of PolyM blocks or polyG blocks (10 mg/mL) in 50 mM phosphate buffer (pH 7.6). The reaction mixture was then incubated at 40 °C for 24 h. The reaction products were analyzed by thin-layer chromatography (TLC) with the unfolding agent (butanol/acetic-acid/water 2:1:1, *v*/*v*) and the saccharides were visualized with a developing reagent (sulfuric acid/ethanol reagent 1:4, *v*/*v*) after heating the TLC plate at 80 °C for 30 min [30]. The products were also analyzed by fast protein liquid chromatography (FPLC) with a Superdex peptide 10/300 gel filtration column (GE Health, Chicago, IL, USA), as previously described [31]. The Dionex software Chromeleon v7.2 (Thermo Scientific, Pittsburgh, PA, USA) was used to identify and quantify sugars. The reaction products were further characterized by negative-ion electrospray ionization mass spectrometry (ESI-MS) [19]. As a control, two types of the acid hydrolyzed saturated alginate oligosaccharides (SAOS) (saturated mannuronate oligosaccharides (SMOS) and saturated guluronate oligosaccharides (SGOS)) were prepared according to the method of Li et al., 2016 [30]. The acid hydrolysis products were also analyzed by TLC. The average molecular weight of these AOS were analyzed by the carbazole sulfate method [19]. Chitooligosaccharides (COS) are the degradation products of chitosanase CsnM [32]. The COS used in this study were also analyzed by TLC and positive-ion ESI-MS (Appendix A). The average molecular weight of COS was 714.06.

### 2.3. Animals and Diet

KM mice were provided by the Darenfucheng Animal Husbandry Company (Qingdao, China). The mice were housed in a specific-pathogen-free (SPF) room (temperature: 21 ± 2 °C, humidity: 45–65%, and 12 h dark–light cycles). Water was given to the mice freely during experiments. After seven days of acclimation (17–18 g), mice were divided into the following ten groups (8 per group): (1) STD: standard treatment diet group, fed a normal diet for 8 weeks, (2) HFD: fed an HFD for 8 weeks, (3) COS: fed an HFD for 4 weeks and then fed an HFD with 350 mg/kg per day and COS for 4 weeks, (4) Orlistat: fed an HFD for 4 weeks and then fed a HFD with 75 mg/kg per day with Orlistat for 4 weeks, (5–7) UAOS: fed an HFD for 4 weeks and then fed an HFD with 100 mg/kg per day (L), 200 mg/kg per day (M) or 400 mg/kg per day (H) UAOM for 4 weeks, (8–10) SAOS: fed an HFD for 4 weeks and then fed an HFD with 100 mg/kg per day (L), 200 mg/kg per day (M) or 400 mg/kg per day (H) SAOS for 4 weeks. Orlistat, COS, UAOS, and SAOS were dissolved in distilled water at a concentration of 600 mg/mL and were then administered at a dose of 1 mL/100 mg daily by gavage. Afterwards, other KM mice (16–18 g) were purchased and housed as above. After 7 days of acclimation, mice were divided into the following seven groups (12 per group): (1) STD: standard treatment diet group, with 8 weeks of a normal diet, (2) HFD: with 8 weeks of HFD, (3) COS: 4 weeks of HFD and 4 weeks of HFD with 350 mg/kg.day COS, (4) UMOS: 4 weeks of HFD and 4 weeks of HFD with 350 mg/kg per day UMOS, (5) UGOM: 4 weeks of HFD and 4 weeks of HFD with 350 mg/kg per day UGOM, (6) SMOS: 4 weeks of HFD and 4 weeks of HFD with 350 mg/kg per day SMOS, (7) SGOS: 4 weeks of HFD and 4 weeks of HFD with 350 mg/kg per day SGOS. COS, UMOS, UGOM, SMOS, and SGOS were dissolved in distilled water at a concentration of 600 mg/ml and were then administered at a dose of 1 mL/100 mg daily by gavage.

In the end of the experiment, the mice were fasted for 12 h and sacrificed. Blood was obtained from the retro-orbital plexus. Serum was obtained via centrifuge (3500 rpm, 4 °C, 30 min). Liver and adipose tissues were quickly stripped and weighed. After a picture was taken, the tissues were stored at –80 °C for further analysis. The animals used in these experiments were approved by the Ethics Committee of the Medical College of Qingdao University.

### 2.4. Calculation of Energy Intake, Weight Gain, and Lee’s Index

Daily food intake was recorded and body weight and body length of each mouse were measured weekly. Lee’s index was calculated from the formula: body weight (g) 1/3 × 1000/body length (cm).

### 2.5. Biochemical Analysis

The concentration of triacylglycerol (TG), total cholesterol (TC), high-density lipoprotein (HDL), and low-density lipoprotein (LDL) in mouse serum was measured using commercial detection kits (Nanjing jiancheng Bioengineering Institute, Nanjing, China). An enzyme-linked immunosorbent assay kit to measure the concentration of free fatty acids (FFA) was purchased from Bio-Techne (Shanghai, China). Hepatic hydrogen peroxide concentration was measured by a hydrogen peroxide testing kit (Beyotime Biotech., Haimen, China). Hepatic lipoperoxide levels were measured using an MDA testing kit (Nanjing Jiancheng Biotec, Nanjing, China).

### 2.6. Measurement of Hepatic Lipids, AST, and ALT

Liver tissue (0.1 g) was homogenized with 0.9 mL sodium chloride (0.9%). The supernatant was then collected via centrifugation (2500 rpm, 4 °C, 10 min). The levels of hepatic TC, TG, AST, and ALT were determined using commercial detection kits (Nanjing jiancheng Bioengineering Institute, Nanjing, China). Hydrogen peroxide generation was assessed as previously described [33], while lipid peroxidation was quantified as malondialdehyde (MDA) and expressed as nanomoles of MDA per milligram of protein [34].

### 2.7. Histological Analysis

Tissues were dissected and washed with saline and then fixed using 10% formalin solution for 24 h. Tissue cutting and hematoxylin and eosin (H&E) staining were performed by Sevicebio (Wuhan, China). Images were obtained using a microscope at 200× magnification. The adipocyte sizes were determined using ImageJ (NIH, Bethesda, MD, USA) (https://imagej.nih.gov/ij/).

### 2.8. mRNA Quantification by Quantitative Real-Time PCR (qPCR)

Total RNA was isolated from tissues and qPCR were performed as previously described [35]. The primers used in this study were synthesized by Sangong Biotech (Shanghai, China) (Appendix A). Gene expression levels were normalized to *GAPDH* and the relative quantification of gene expression was calculated using the 2^−∆∆Ct^ method.

### 2.9. Western Blotting

Total protein content was obtained from tissue lysis using cold RIPA lysis buffer, which was followed by centrifugation at 12,000 g for 30 min at 4 °C. This process was repeated three times. The BCA protein assay kit was used to quantify the harvested protein. Western blotting was performed as previously described [35].

### 2.10. Statistical Analysis

All images in this study were formatted for optimal presentation using Adobe Illustrator CS4 (Adobe Systems, Inc., San Jose, CA, USA). To determine the statistical significance between two groups, a Student’s t-test was performed to calculate the associated P-values. The statistical significance between multiple groups was calculated by one-way analysis of variance (ANOVA) using the GraphPad Prism 5 (GraphPad Software, La Jolla, CA, USA).

## 3. Results

### 3.1. Separation and Determination of AOS

Two types of AOS were prepared in this study, including enzymatic UAOS and acid hydrolyzed SAOS. UAOS was performed using a green production method developed in our previous study, by combining enzymatic hydrolysis and selective fermentation directly from LJ [19]. The scheme of UAOS and SAOS was shown in Figure 1A. The structural difference between them is that UAOS contains an unsaturated double bond between C4 and C5. Alginate is a linear polysaccharide containing M block and its C5 epimer G block. To further identify any differences in activity between M and G fragments, UAOS were separated into UMOS and UGOS. Meanwhile, SAOS were also separated into SMOS and SGOS as a control. According to the carbazole sulfate analysis, the average molecular weights of these four AOS were similar (UMOS, 446.5, UGOS, 441.6, SMOS, 459.6, SGOS, 455.2). To further analyze the consistency of these AOS, TLC analysis was performed. The mobility of the separated UMOS, UGOS, SMOS, and SGOS were similar (Figure 1B).

In order to analysis the main ingredient of the separated UAOS, UMOS and UGOS were measured by SE-HPLC using the Superdex peptide 10/300^TM^ column. As shown in Figure 1C,D, the main ingredients are disaccharides (DP2) and trisaccharides (DP3). Negative ion ESI-MS was also used to further determine the molecular weight of UMOS and UGOS. As shown in Figure 1E,F the main peaks at 351.03 m/z and 527.04 *m/z* corresponded to unsaturated alginate disaccharides and unsaturated alginate trisaccharides. These results indicated that the main ingredients of UMOS and UGOS used in this study are dimers and trimers, which are in agreement with its average molecular weights.

### 3.2. Effects of AOS on Body Weight

To evaluate the anti-obesity effects of two types of AOS, the high-fat diet obese model was established. Orlistat is the only over-the-counter drug approved by the Food and Drug Administration (FDA) to treat obesity and used as a positive control [7]. In addition, COS was also prepared and used as a positive control. According to the TLC and ESI-MS analysis, the main ingredients of COS are dimers, trimers, and tetramers (Appendix A). As shown in Figure 2A–C, the effect of COS on reducing body weight gain for HFD-induced obesity was similar to Orlistat. Several studies have shown that COS has an anti-obesity effect and less side effects [36]. Therefore, COS was used as a positive control to verify the anti-obesity effects of AOS, which are also functional oligosaccharides. UAOS showed a dose-dependent effect on weight loss. However, SAOS did not have a similar effect on weight loss.

To verify whether structural differences caused the different anti-obesity effects, we selected the four different types of AOS, including UAOS (UMOS and UGOS) and SAOS (SMOS and SGOS) for further studies. To assess the anti-obesity effects of AOS, the energy intake and body weight of mice were measured weekly. As shown in Figure 2D, the HFD group had slightly higher energy intake than the standard treatment diet group (STD), but the difference was not significant. The HFD foods contain more calories and this indicated that there was no influence of AOS or COS treatment on the appetite of the mice. As shown in Figure 2E, following four weeks of a high-fat diet feeding and an additional four weeks of COS or four types of AOS treatment, there was a significant difference in body weight between the STD and HFD groups. In terms of body weight gain, significant effects were seen after four weeks of UAOS treatment (UMOS: *p* < 0.001 and UGOM: *p* < 0.01) and COS treatment (*P* < 0.001). UGOM was a little less effective than UMOS (*P* < 0.01) and MOS and GOS did not show clear effects (Figure 2F). In addition, there were significant differences between COS, UMOS, and UGOM treatment groups and the HFD/STD groups. The acid hydrolyzed SAOS (MOS and GOS) treatment groups showed no difference (Figure 2G). These results suggested only that the UAOS has an effective weight-loss effect in HFD mice. Meanwhile, there was no significant difference in the effects of UMOS and UGOS, which indicated that the structural difference of M and G in the C5 epimer is not the main factor in UAOS-induced alteration of body weight.

### 3.3. Effect of UAOS on Serum Lipids

In order to determine the effect of different AOS on serum lipids, four different UAOS were used and the results were analyzed, as shown in Figure 3. Compared with the STD group, there was a significant increase in TG, TC, and FFA levels, as well as a significant decrease of HDL-c, but no significant difference in LDL-c, in the HFD group. The results indicated that the HFD mice had symptoms of hyperlipidemia. The positive control, COS, clearly lowered serum TG and TC levels. Moreover, compared with SAOS (SMOS and SGOS), UAOS (UMOS and UGOM) treatment markedly lowered serum TG and TC, and decreased FFA levels. These results suggest that the enzymatic UAOS could reduce blood lipids in HFD-induced obese mice.

### 3.4. Effects of UAOS on Liver Protection

To evaluate the effect of different AOS on liver protection, the effects of UAOS on the liver were examined after eight weeks of treatment. Compared with the STD group, the liver weight (Figure 4A) of HFD-fed mice were significantly increased. Treatment with COS, UMOS (*p* < 0.01), or UGOM (*P* < 0.01) reduced HFD feeding induced liver weight. For the concentration of liver TC (Figure 4B) and TG (Figure 4C), significant increases in the HFD group were shown when compared with the STD group. This indicated that HFD feeding increased liver lipid accumulation. After treatment with UAOS (UMOS an UGOM), TC and TG levels induced in HFD feeding in the liver were significantly reduced. Moreover, the alteration of liver weight (*p* < 0.05), liver TC (UAOS and SAOS), and TG (UAOS and SAOS) caused by UAOS and SAOS treatment were significantly different (Figure 4A–C).

According to the liver morphology (Figure 4D), whole livers in the HFD group were much bigger than the STD group, which suggested that fatty liver had developed. According to the liver H&E staining (Figure 4E), there were large histological abnormalities of hepatocytes in the HFD group with large fat vacuoles. This indicates that the mice had suffered a high degree of hepatic steatosis induced by the HFD. COS and UAOS markedly decreased the liver fat and fat vacuole size induced by HFD. Liver injury and hepatotoxicity are the two main factors for hyperlipidemia and obesity [37,38]. To further check the hepatoprotective effect of UAOS, serum AST and ALT levels were measured (Figure 4F,G). Serum AST and ALT levels were significantly increased after HFD when compared to the STD diet (*P* < 0.001). UAOS (UMOS and UGOM) significantly lowered the concentration of ALT and AST in serum. The effects of UAOS (especially UMOS) were similar to COS, which suggests that UAOS play an important role in liver protection. These results demonstrated that UAOS instead of SAOS exert a liver protective effect by reducing fat accumulation and relieving fatty liver.

To further understand the molecular mechanisms of UAOS liver protection effects, we used qPCR to measure gene expression levels of specific genes, specifically *SREBP-1c*, *FAS*, *ACC*, and *HMGCR*, involved in liver adipogenesis, lipogenesis, and lipolysis. In the obesity model, *SREBP-1c*, *FAS*, *ACC*, and *HMGCR* were increased while adiponectin was decreased in liver. Significant increases of *SREBP-1c*, *FAS*, *ACC*, and *HMGCR* and a significant decrease in adiponectin in hepatic tissue were observed in the HFD group (Appendix A). Treatment with COS and UAOS, especially UMOS, reversed the upregulated expression of *SREBP-1c*, *FAS*, *ACC*, and *HMGCR*. The downregulated expression of adiponectin was induced by HFD feeding. The results indicated that UAOS plays a critical role in hepatic lipid metabolism.

### 3.5. Effects of UAOS on Adipose Tissue Mass

The effects of UAOS (UMOS and UGOM) on adipose tissues were also examined after the eight-week administration (Figure 5). Compared with the STD fed mice, the weights of epididymal (Figure 5A), mesenteric (Figure 5B), perirenal white adipose tissues (WAT) (Figure 5C), and adipocyte size (Figure 5D) in the HFD group were markedly increased. In addition, UAOS were found to reduce the weights of epididymal, mesenteric, perirenal WAT and adipocyte size caused by an HFD. Moreover, the inhibitory effects of UMOS on HFD-induced elevated adipose tissue mass were similar to that of COS. Moreover, according to the morphology of epididymal WAT, the size of WAT in HFD mice was higher than in the STD group (Figure 5E) and, according to the H&E staining of WAT, the size of adipocytes from HFD mice were bigger than in the STD group (Figure 5F). Treatment with COS and the two UAOS (UMOS and UGOM) inhibited the growth of adipocytes and the size of adipose tissue, since smaller epididymal WAT and smaller adipocytes were observed when compared to mice in the HFD group (Figure 5E,F). Therefore, UAOS can play an effective anti-obesity role by inhibiting the growth of adipocytes.

WAT is a type of adipose tissue whose main function is to store excess fat in the body, which leads to obesity. Hyperplasia and hypertrophy of WAT can lead to obesity and metabolic syndrome [39,40]. Therefore, inhibiting the formation of WAT is an effective strategy for the treatment of obesity. In the obesity model, *PPARγ*, *C/EBPα*, *STREBP-1c*, *ACC*, *FABP4*, *FAS*, and *HSL* were increased while *PLIN* was decreased in WAT. In this study, we also measured gene expression levels of specific genes including *PPARγ*, *C/EBPα*, *STREBP-1c*, *ACC*, *FABP4*, *FAS*, *HSL*, and *PLIN* in WAT. Significant increases in *PPARγ*, *C/EBPα*, *STREBP-1c*, *ACC*, *FABP4*, *FAS*, and *PLIN* and a significant decrease in HSL expression in WAT tissue was observed in the HFD group (Appendix A). Treatment with COS and UAOS, especially UMOS, reversed the upregulated expression of *PPARγ*, *STREBP-1c*, *ACC*, *FABP4*, *FAS*, and *PLIN*, and the downregulated expression of *HSL* was induced by HFD feeding. However, there was no effect on the expression of *C/EBPα* following COS or UAOS administration. These results indicated that UAOS plays a critical role in lipid accumulation in epididymal WAT. The above results also showed that changes in gene expression related to lipid metabolism in the liver and WAT suggest that UAOS can inhibit adipogenesis in the liver and WAT to reduce obesity.

### 3.6. Effects of UAOS on Liver Hydrogen Peroxide and Malondialdehyde Level

In this study, UAOS showed excellent hepatoprotective and anti-obesity effects, but acid hydrolyzed SAOS showed no effect on HFD-fed mice. Since the structural difference between UAOS and SAOS is an unsaturated bond, we further tested their antioxidant capabilities. An HFD increased hepatic generation of hydrogen peroxide, as well as increasing hepatic ROS levels compared with the STD diet group (Figure 6). Compared with the HFD control group, treatment with UAOS significantly lowered hepatic hydrogen peroxide (UMOS: 42.1% and UGOM: 25.3%, Figure 6A) and malondialdehyde (MDA) levels (UMOS: 28.5% and UGOM: 24.7%, Figure 6B). Moreover, the difference between UAOS (UMOS and UGOM) and SAOS (SMOS and SGOS) caused an alteration of liver hydrogen peroxide (*p* < 0.05). Liver MDA (*p* < 0.01) were significant. These results indicate that UAOS have the potential to alleviate HFD-induced ROS formation and accumulation.

### 3.7. Effects of UAOS on AMPK Signaling

AMPK is an important cellular energy sensor that can regulate energy metabolism. AMPKα is the main catalytic subunit in WAT tissue and enhances lipid metabolism after phosphorylation on Thr-172 [41,42,43]. Therefore, the activation of AMPK and its downstream signaling protein ACC in WAT tissues were detected using Western blotting (Figure 7A,B). Protein expression levels of p-AMPK and p-ACC in the HFD group were significantly decreased compared to the STD group. Treatment with UAOS significantly reversed the inactivation of AMPK and ACC (especially UMOS). These results suggested that UAOS (especially UMOS) administration could improve obesity by activating AMPK and the ACC signaling pathway to enhance lipid catabolism. Based on these results, we have proposed a mechanism by which UAOS causes effects of hepatoprotection and anti-obesity (Figure 7C).

## 4. Discussion

Obesity and its co-morbidities seriously threaten human health [44]. Since existing anti-obesity drugs have strong side effects, the development of functional foods as adjuvant treatments is urgently needed [6,45]. In this study, we investigated the use of a non-toxic green food additive, UAOS, from the enzymatic degradation of LJ, which showed effective anti-obesity effects in an HFD-mouse model. As documented previously, AOS have been reported to have various biological functions, such as anti-oxidant activity, immunomodulatory effect, and anti-tumor activity [45,46,47,48]. However, to the best of our knowledge, this is the first report about anti-obesity activity of AOS. The structural differences are often accompanied by changes in biological activity [49]. Subdivide the structures of AOS components. Four types of AOS are determined, including acid hydrolyzed SAOS (SMOS and SGOS) and enzymatic UAOS (UMOS and UGOS). M and G are epimers in C5. Our results indicate that the structural difference of M and G does not result in significantly different functions in terms of hepatoprotective and anti-obesity activity (Figure 2, Figure 3, Figure 4 and Figure 5). This study indicated that, compared with SAOS, the UAOS showed a significantly anti-obesity effect (Figure 2). The discovery of these excellent biological activities of enzymatic UAOS will promote the development of green processing methods for brown seaweeds.

In a previous study, compared with SAOS, the non-reducing ends of UAOS provide double bonds between C4 and C5, giving them greater *in vitro* anti-oxidant activity than ascorbic acid in lipid oxidation treatment [46]. This study showed that UAOS also played effective anti-oxidant activity in vivo tissue (Figure 6). This excellent characteristics make UAOS reduce excessive ROS accumulation, which maintains mitochondrial homeostasis. Obesity leads to metabolic stress resulting in liver injury, including non-alcoholic fatty liver disease (NAFLD). Therefore, efforts to reduce fat mass for preventing and treating NAFLD have been sought [50,51]. Our results indicate that UAOS treatment significantly reduced the concentration of ALT and AST in serum and played a role in liver protection by reducing fat accumulation and relieving fatty liver. In NAFLD, oxidative stress and dysfunction of mitochondria are linked with disorders in β-oxidation of free fatty acids [52,53]. Excessive liver ROS production and decreased cellular antioxidant activity lead to oxidative stress and liver oxidative damage [21]. In addition, it has shown that obese, high glucose-fed rats showed increases in plasma levels of triglycerides, superoxide anion production, and NADPH oxidase activity in the adipose tissue [54]. Moreover, argan oil as an antioxidant diet was found to reduce the HFD or high glucose diet, which induced an increase in adipose tissue weight, body weight, and serum triglyceride concentrations. This suggests that adipose tissue oxidative stress is partly implicated in developing visceral obesity [54,55]. Therefore, antioxidants can effectively improve lipid and ROS metabolism, which, thereby, reduces liver steatosis and oxidative damage. This study also showed that UAOS have better hepatoprotective activity than SAOS (Figure 4), which corresponds to their anti-oxidant activity. However, due to increased adipose tissue leading to obesity, whether HFD induced oxidative stress is related to obesity development in our animal model. If AOSs exert beneficial impacts on obesity through their effects on adipose tissue oxidative stress, which need to be further studied.

Fat accumulation in adipose tissues induced by a high-fat diet is one of the causes of obesity. Adipocytes play an important role in energy regulation and energy metabolism. Our results showed that UAOS significantly reduced the growth of adipocytes, which suggests excellent anti-obesity effects. Simultaneously, UAOS also reduced serum TG, TC, and FFA levels caused by high fat diets, which also contribute to its anti-obesity effects. AMPK is an important energy sensor that modulates energy balance and lipid metabolism. Our results showed that UAOS treatment increased AMPK activation (Figure 7). Therefore, the anti-obesity effect of UAOS works by regulating the AMPK signaling pathway. Several studies have reported that AMPK plays a central role in regulating glucose and lipid metabolism. It has previously been reported that AMPK activation inhibits lipogenesis and adipocyte differentiation. Furthermore, ACC as downstream of AMPK is a multi-subunit enzyme that regulates enzymes involved in malonyl-CoA production, fatty acid synthesis, and fatty acid oxidation in adipocyte [56,57].

## 5. Conclusions

In this study, UAOS as a non-toxic green food showed effective anti-obesity effects in an HFD-fed mice model. UAOS could significantly increase both AMPK and ACC phosphorylation in adipocytes, which indicated that UAOS played an anti-obesity effect through the AMPK signaling pathway. These results make UAOS a good candidate for adjuvant therapy in anti-obesity treatment. Further work will focus on the molecular mechanism in cell lines and the gut microbiota change met of its anti-obesity activity.

## Figures and Tables

**Figure 1 marinedrugs-17-00540-f001:**
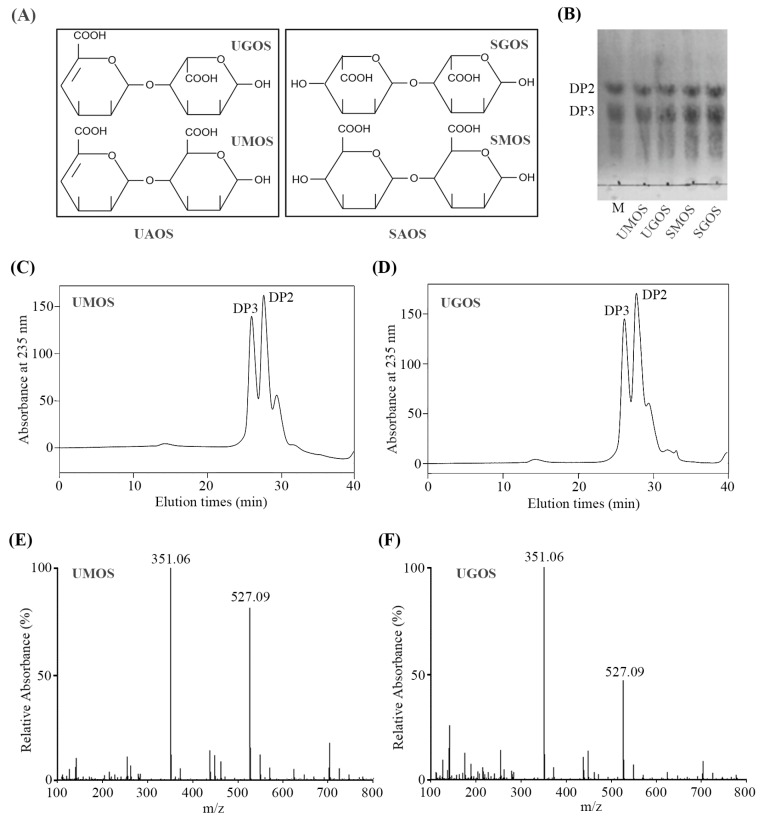
Scheme (**A**), TLC analysis (**B**), SE-HPLC analysis (**C**,**D**), and ESI-MS analysis (**E**,**F**) of UAOS and SAOS.

**Figure 2 marinedrugs-17-00540-f002:**
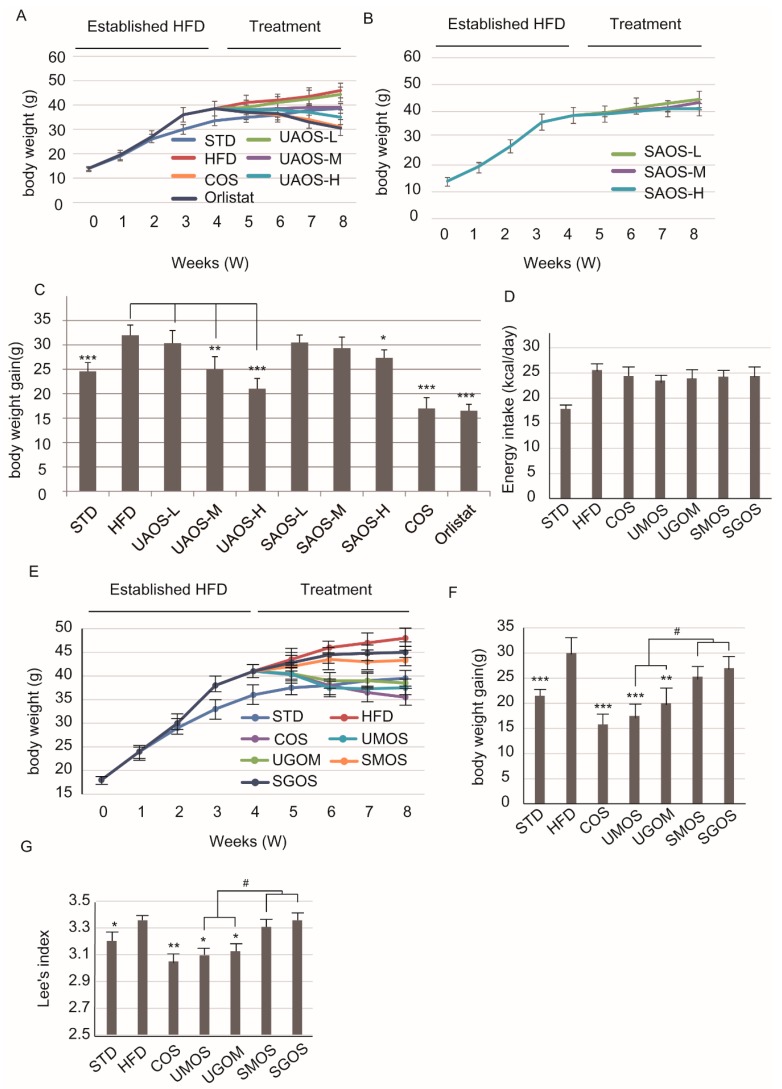
Changes in body weight on AOS treatment. Changes in body weight of treatment of UAOS (**A**) and SAOS (**B**), body weight gain (**C**) for 4 weeks. The data are represented as means ± standard deviation (SD, *n* = 6). Changes in the energy intake (**D**), body weight (**E**), body weight gain (**F**), and Lee’s index (**G**) during the eight-week treatment are shown. The body weight was recorded weekly. The data are represented as means ± standard deviation (SD, *n* = 12). Compare with HFD group, * *p* < 0.05, ** *p* < 0.01, *** *p* < 0.001. Compare with indicated groups, ^#^
*p* < 0.05.

**Figure 3 marinedrugs-17-00540-f003:**
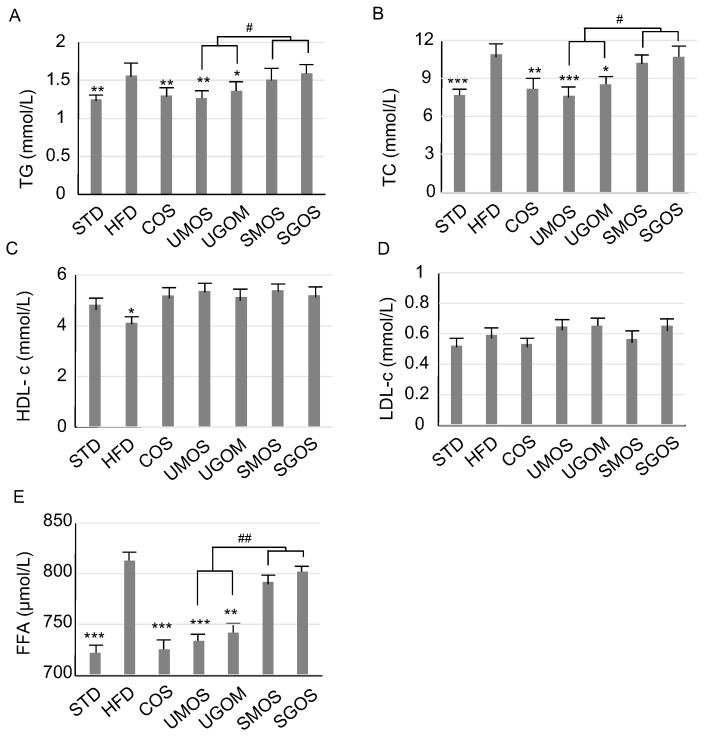
Changes of AOS treatment on serum lipid. TG levels (**A**), TC levels (**B**), HDL-c levels (**C**), LDL-c levels (**D**), and FFA levels (**E**). The data are represented as means ± standard deviation (SD, *n* = 12). Compare with the HFD group, * *p* < 0.05, ** *p* < 0.01, *** *p* < 0.001. Compare with indicated groups, ^#^
*p* < 0.05, ^##^
*p* < 0.01.

**Figure 4 marinedrugs-17-00540-f004:**
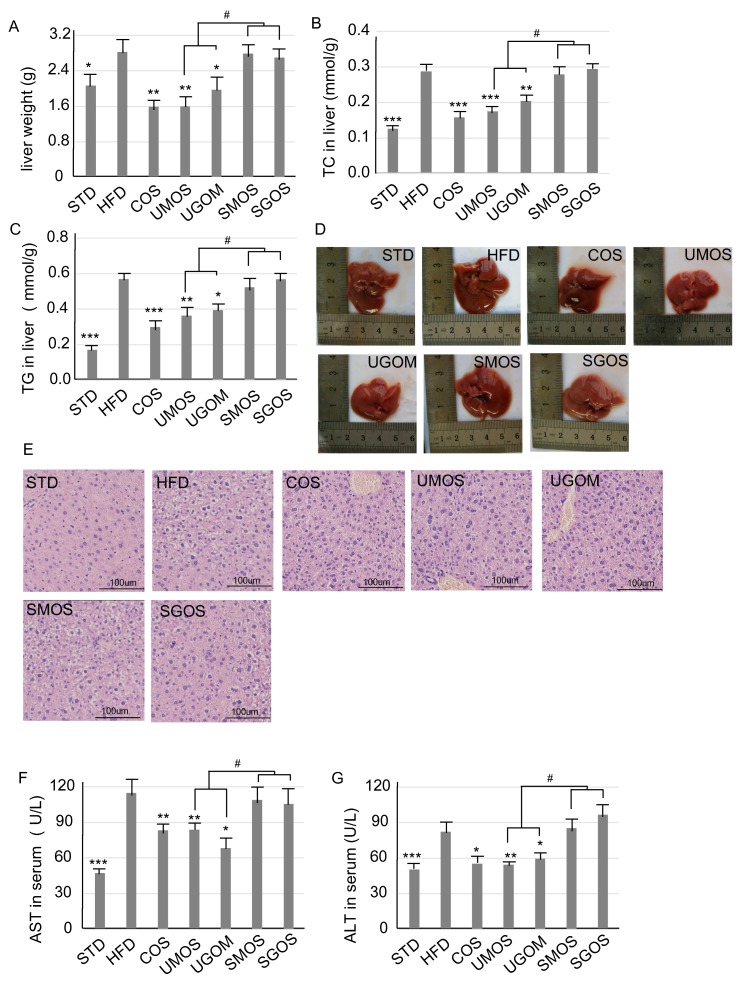
Changes of AOS treatment on the hepatic lipid. The whole liver weight (**A**), liver TC (**B**), and TG (**C**) are displayed. The whole liver morphology (**D**) and H&E staining for liver tissue (**E**) (200×) are shown. The concentration of serum AST (**F**) and ALT (**G**) are also shown. The data are represented as means ± standard deviation (SD, *n* = 12). Compare with the HFD group, * *p* < 0.05, ** *p* < 0.01, *** *p* < 0.001. Compare with indicated groups, ^#^
*p* < 0.05.

**Figure 5 marinedrugs-17-00540-f005:**
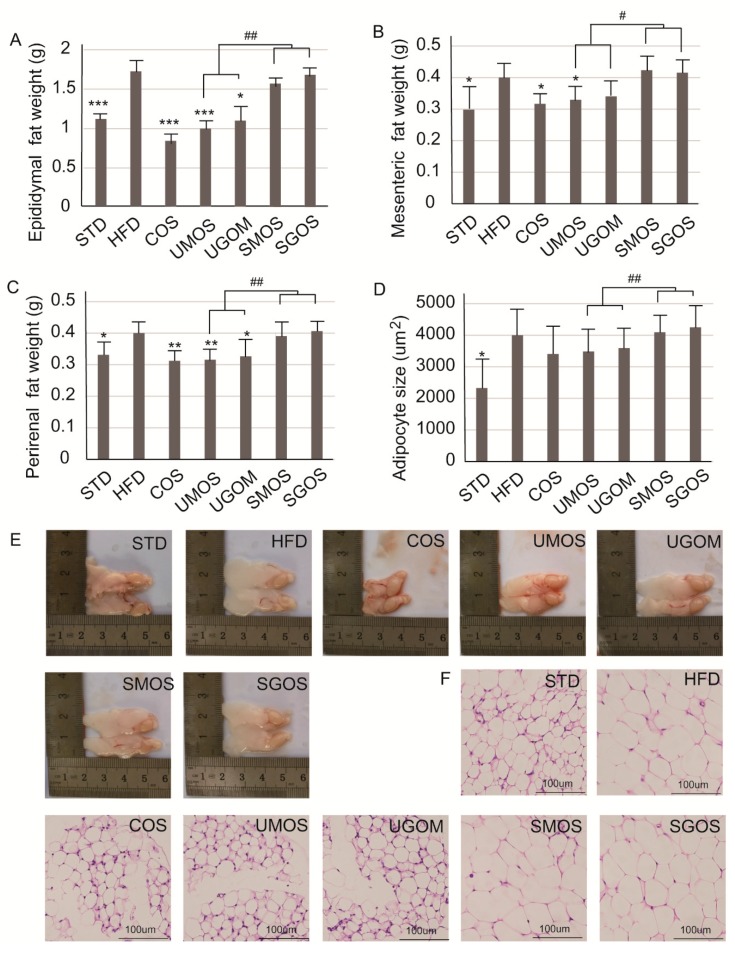
Changes of UAOS treatment on adipocytes in adipose mass. The fat mass of epididymal (**A**), mesenteric (**B**), perirenal tissues (**C**), adipocyte size (**D**), the whole WAT (**E**), and WAT tissue stained by H&E staining (200×) (**F**) are shown in the figure. The data are represented as means ± standard deviation (SD, *n* = 12). Compare with the HFD group, * *p* < 0.05, ** *p* < 0.01, *** *p* < 0.001. Compare with indicated groups, ^#^
*p* < 0.05, ^##^
*p* < 0.01.

**Figure 6 marinedrugs-17-00540-f006:**
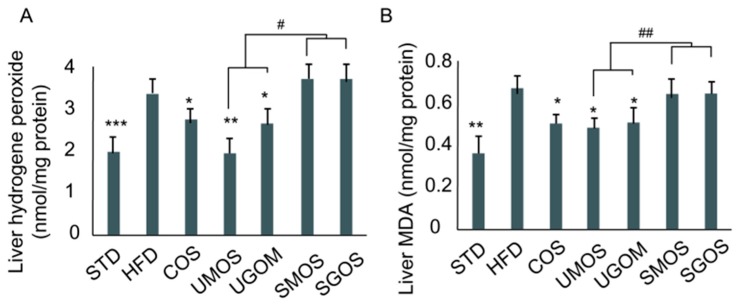
The effect of AOS treatment on hepatic H_2_O_2_ level (**A**) and liver MDA level (**B**) in mice fed a high fat diet for 8 weeks. The data are represented as means ± standard deviation (SD, *n* = 12). Compare with the HFD group, * *p* < 0.05, ** *p* < 0.01, *** *p* < 0.001. Compare with indicated groups, ^#^
*p* < 0.05, ^##^
*p* < 0.01.

**Figure 7 marinedrugs-17-00540-f007:**
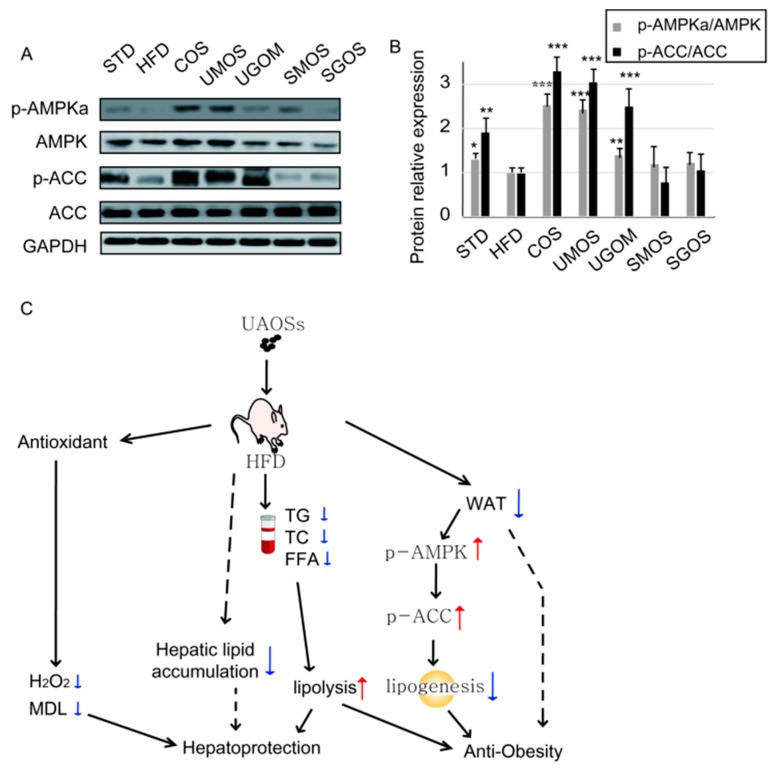
UAOS activated the AMPK signaling pathway in WAT. (**A**) The protein expression level of AMPK, p-AMPK, ACC, and p-ACC in WAT were detected by Western blotting. (**B**) The relative protein levels were quantified. (**C**) The proposed mechanism by which AOS causes effects of hepatoprotection and anti-obesity. The data are represented as means ± standard deviation (SD, *n* = 3). * *p* < 0.05, ** *p* < 0.01, *** *p* < 0.001.

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
