# Peer review of "Efficiently Anti-Obesity Effects of Unsaturated Alginate Oligosaccharides (UAOS) in High-Fat Diet (HFD)-Fed Mice"

_marinedrugs, 2019, doi:10.3390/md17090540_

Round 1

Reviewer 1 Report

The authors investigated the effects of unsaturated alginate oligosaccharides on body weight, lipid profile, hepatic oxidative stress and AMPK signaling pathway in high-fat diet fed mouse, an animal model of obesity. This study is of interest however, some concerns should be addressed.

Major comments

The results of the present study are very interesting. However, the manuscript is not well written.  The results are unfortunately poorly discussed. They need to be more challenged in the discussion section in order to support the conclusions.

In the results section,  page 5, line 203, the authors stated “As shown in Figure 2A and 2B, the effect of COS on reducing body weight gain for HFD-induced obesity was similar to Orlistat”, while in page 6, figure 2A and 2B, the authors presented only the body weight and not the body weight gain parameter. This latter would reflect more adequately the effects of AOSs on obesity. Moreover, in the same figures, the significance differences between groups are not appropriately presented.

In order to better evaluate the effects of AOSs treatment on the liver weight, the authors should present the figure 4A as liver weight per body weight. Also, the effects of UAOS treatment on adipose mass would be evaluated more adequately by presenting the figures 5A, 5B and 5C as epididymal, mesenteric, peri-renal fat weights per body weight.

There are many comparisons between groups which appear to be not significant. I will give two examples: the difference in adipocyte size between HFD and control animals or COS group observed in figure 5D seems to be not significant as there is a high variation in each group. The authors should check the accuracy of these comparisons and others.

The authors did evaluate the effects of AOSs on obesity and lipid profile. As insulin resistance is associated with obesity, this study would be more interesting if they include the results of plasma glucose and insulin levels as well as insulin resistance index.

What is the impact of oxidative stress produced at the level of adipose tissue in the development of obesity in this animal model? Could AOSs exert beneficial impacts on obesity through their effects on adipose tissue oxidative stress?

The authors concluded that UAOS had an anti-obesity effect mainly through AMPK signaling. However, in the introduction section, the authors did not introduce the potential implication of such signaling pathway in the development of obesity. Moreover, several paragraphs in the results section should be moved to results or discussion sections appropriately.

Minor comments

In line 73, “UAOS may be promising candidates for the treatment of metabolic diseases such as fatty liver, hypertriglyceridemia and possibly diabetes” should be replaced by “UAOS may be promising candidates for the treatment of metabolic diseases such as fatty liver, hypertriglyceridemia and possibly type 2 diabetes”.

In line 240, the authors mentioned “Moreover, compared with SAOS (SMOS and SGOS), UAOS (UMOS and UGOM) treatment markedly lowered serum TG and TC, with higher FFA levels”. This sentence should be corrected.

In line 256, the authors stated “Moreover, the difference between UAOS and SAOS caused alteration of liver weigh, liver TC and TG is significantly”. This sentence should be corrected.

The legend of figure 5 should be corrected.

In line 301, “metabolic syndromes” should be replaced by “metabolic syndrome”.  

Author Response

Response to Reviewer 1 Comments

The authors investigated the effects of unsaturated alginate oligosaccharides on body weight, lipid profile, hepatic oxidative stress and AMPK signaling pathway in high-fat diet fed mouse, an animal model of obesity. This study is of interest however, some concerns should be addressed.

Major comments

Point 1: The results of the present study are very interesting. However, the manuscript is not well written.  The results are unfortunately poorly discussed. They need to be more challenged in the discussion section in order to support the conclusions.

Response 1: Thank you very much for your kind works. In the revised manuscript, we have rewritten the discussion section to make the manuscript more logical.

The details as following:

In the first part, we focus on the contribution of the first finding towards the anti-obesity activity of AOS.  Even AOS have been reported to have various biological functions, such as anti-oxidant activity, immunomodulatory effect, and anti-tumor activity, this is the first report of its anti-obesity activity.

In the second part, we discuss the effect of structural differences on its biological activity. Our results indicate that the structural difference of M and G does not result in significantly different functions in terms of hepatoprotective and anti-obesity activity. Interestingly, this study indicated that compared with SAOS, the UAOS showed significantly anti-obesity effect.

In the third part, we discuss the anti-oxidant activity. The difference between SAOS and UAOS are the anti-oxidant. In previous study, compared with SAOS, the non-reducing ends of UAOS provide double bonds between C4 and C5, giving them greater in vitro anti-oxidant activity than ascorbic acid in lipid oxidation treatment. This study showed that UAOS also played effective anti-oxidant activity in vivo tissue. This excellent characterises make UAOS reducing excessive ROS accumulation, thereby maintaining mitochondrial homeostasis. Obesity leads to metabolic stress resulting in liver injury, including non-alcoholic fatty liver disease (NAFLD). Therefore, efforts to reduce fat mass for the prevention and treatment of NAFLD have been sought.

In the final part, we focus on the effect of UAOS on adipose tissues. Adipocytes play an important role in energy regulation and energy metabolism. Our results showed that UAOS treatment increased AMPK activation. Therefore, the anti-obesity effect of UAOS works by regulation of the AMPK signaling pathway.

Point 2: In the results section,  page 5, line 203, the authors stated “As shown in Figure 2A and 2B, the effect of COS on reducing body weight gain for HFD-induced obesity was similar to Orlistat”, while in page 6, figure 2A and 2B, the authors presented only the body weight and not the body weight gain parameter. This latter would reflect more adequately the effects of AOSs on obesity. Moreover, in the same figures, the significance differences between groups are not appropriately presented.

Response 2: Thank you very much for your kind works. We have overlooked such a concluding result. We added body weight gain data for Figure 2A and 2B. Please see the added result in Figure 2C. We wish the data could be better expressed in this version.

Point 3: In order to better evaluate the effects of AOSs treatment on the liver weight, the authors should present the figure 4A as liver weight per body weight. Also, the effects of UAOS treatment on adipose mass would be evaluated more adequately by presenting the figures 5A, 5B and 5C as epididymal, mesenteric, peri-renal fat weights per body weight.

Response 3: The opinion of reviewer is also absolutely right. Body weight and body length are also factors need to be considered in obesity models.  However, absolute weight is used in most literatures but not the ratio of liver or fat weight and body weight (Cho et al., 2018; Pan et al., 2018). In this study, we have provided lee’s index data (body weight (g) 1/3 × 1000/body length (cm)) in Figure 2F. If other researchers are interested in our data, they can get the desired results from absolute weight and lee’s index. Thank you again.

Point 4: There are many comparisons between groups which appear to be not significant. I will give two examples: the difference in adipocyte size between HFD and control animals or COS group observed in figure 5D seems to be not significant as there is a high variation in each group. The authors should check the accuracy of these comparisons and others.

Response 4: I am sorry for the mistakes. As the suggestion of reviewer, we have re-analyzed all the analysis in the revised manuscript. We also modified the data description in the main text.

Point 5: The authors did evaluate the effects of AOSs on obesity and lipid profile. As insulin resistance is associated with obesity, this study would be more interesting if they include the results of plasma glucose and insulin levels as well as insulin resistance index.

Response 5:  The opinion of reviewer is also absolutely right and we quietly agree. HFD induced obesity and the related metabolic disorders including hyperinsulinemia, hyperglycemia, hepatic steatosis, insulin resistance and glucose intolerance. If the plasma glucose and insulin levels were tested, it will support our conclusion even more. We will add those data in our further study. We are trying to study the mechanism of AOS more deeply, such as at cellular level and intestinal flora. Thank you your suggestions again.

Point 6: What is the impact of oxidative stress produced at the level of adipose tissue in the development of obesity in this animal model? Could AOSs exert beneficial impacts on obesity through their effects on adipose tissue oxidative stress?

Response 6: Thank you very much for your kind works. It has been proved that HFD diet can increase the production of reactive oxygen species (ROS) in liver, leads to the accumulation of lipids in hepatocytes, and thus damage the liver. So, in this study, we studied the changes of reactive oxygen species (ROS) in liver tissue after HFD treatment and UAOS treatment. In addition, there are few articles focus on the relationship of adipose tissue and ROS, so we only studied the changes of ROS in liver tissue, not in adipose tissue in this study. The idea ROS in adipose tissue of is very interesting, we will observe the changes of ROS in fat adipose tissue in our further study.

Point 7: The authors concluded that UAOS had an anti-obesity effect mainly through AMPK signaling. However, in the introduction section, the authors did not introduce the potential implication of such signaling pathway in the development of obesity. Moreover, several paragraphs in the results section should be moved to results or discussion sections appropriately.

Response 7: Thank you very much for your kind works. We have added more explaining about AMPK singling pathway in introduction section. And we re-organized result and discussion sections. The added part as following:

“Obesity is one of a disorder disease which is related to energy imbalance. AMP-activated protein kinase (AMPK) is a crucial cellular energy sensor. Once AMPK activated, it triggers catalytic processes to generate ATP while inhibiting anabolic processes that consume ATP in an attempt to restore cellular energy homeostasis. AMPK is considered as a potential target for the treatment of metabolic disorders (Herzig and Shaw, 2018). Increasing evidence has demonstrated that AMPK can inhibit adipogenesis and suppress the expression of SREBP-1c, PPARγ, and FAS in adipocytes (Day et al., 2017). Therefore In this study, we will study whether AMPK is involved in the anti-obesity mechanism of UAOS.”

Minor comments

Point 8: In line 73, “UAOS may be promising candidates for the treatment of metabolic diseases such as fatty liver, hypertriglyceridemia and possibly diabetes” should be replaced by “UAOS may be promising candidates for the treatment of metabolic diseases such as fatty liver, hypertriglyceridemia and possibly type 2 diabetes”.

Response 8: Thanks for your suggestion. We have corrected the wrong expression in main text as following: “UAOS may be promising candidates for the treatment of metabolic diseases such as fatty liver, hypertriglyceridemia and possibly type 2 diabetes”

Point 9: In line 240, the authors mentioned “Moreover, compared with SAOS (SMOS and SGOS), UAOS (UMOS and UGOM) treatment markedly lowered serum TG and TC, with higher FFA levels”. This sentence should be corrected.

Response 9: Thanks for your comment. We have corrected the expression in main text as following: “Moreover, compared with SAOS (SMOS and SGOS), UAOS (UMOS and UGOM) treatment markedly lowered serum TG and TC, and increased FFA levels”.

Point 10: In line 256, the authors stated “Moreover, the difference between UAOS and SAOS caused alteration of liver weigh, liver TC and TG is significantly”. This sentence should be corrected.

Response 10: Thanks for your comment. We have corrected the expression in main text as following: “Moreover, the alteration of liver weight (p<0.05), liver TC (UAOS and SAOS) and TG (UAOS and SAOS) caused by UAOS and SAOS treatment were significantly different (Figure 4A-4C)”.

Point 11: The legend of figure 5 should be corrected.

Response 11: Thanks for your comment. We have changed the mistakes of the legends of Figure 5.

Point 12: In line 301, “metabolic syndromes” should be replaced by “metabolic syndrome”. 

Response 12: I am very sorry for the mistake and we have changed it in main text.

Reviewer 2 Report

Comments

1) Title: oligosaccharides (UAOS) but not oligosaccharide (UAOS).

2) Different abbreviations may make the readers confuse, such as UAOs, UAOS, and UAOSs, as well as AOS and AOSs.

3) Materials and Methods: The authors should indicate the nutritional composition of high-fat diet (HFD) and standard diet (STD), especially energy (kcal) and fat kcal (%).

4) Furthermore, the authors should provide the determination method of adipocyte sizes.

5) If the authors have the permission number of Qingdao University for animal experiments, it should be indicated.

6) Genes should be indicated in italics.

7) The authors should explain the relationship between obesity and the used genes (PPARγ, C/EBPα, STREBP-1c, ACC, FABP4, FAS, HSL, and PLIN) in more detail. Are these genes increased or decreased in obesity?

8) Figure 1E and F) Average molecular weights show UAOS > SAOS (page 4, line 183-184). However, the molecular weight of UAOS and SAOS determined by ESI-MS analysis exhibits the same m/z. It is a mistake?

9) Figure 2 and 4: The mark (##) is not found.

10) Figure 5D: adipocyte size but not adipocytes size.

11) References: The authors should rewrite references, for example see refs. 5, 19, 27, 34, etc.

Author Response

Response to Reviewer 2 Comments

Point 1: Title: oligosaccharides (UAOS) but not oligosaccharide (UAOS).

Response 1: Thank you very much for your suggestion. We have modified manuscript title.

Point 2: Different abbreviations may make the readers confuse, such as UAOs, UAOS, and UAOSs, as well as AOS and AOSs.

Response 2: Thanks for your comment. We have modified the main text using consistent expression. UAOs is a typo mistake, we have corrected as UAOS. We used UAOS and AOS as final expression across all the main text.

Point 3: Materials and Methods: The authors should indicate the nutritional composition of high-fat diet (HFD) and standard diet (STD), especially energy (kcal) and fat kcal (%).

Response 3: Thanks for your comment. We have added ingredient composition of the experimental diets of HFD and STD in Table S1 which including the Macronutrient and ingredient composition of STD and HFD diet.

Point 4: Furthermore, the authors should provide the determination method of adipocyte sizes.

Response 4: Thanks for your comment. We have added the determination method of adipocyte sizes in materials and methods section as following “The adipocyte sizes were determined using ImageJ (NIH, Bethesda, MD, USA) (https://imagej.nih.gov/ij/)”.

Point 5: If the authors have the permission number of Qingdao University for animal experiments, it should be indicated.

Response 5: Thanks for your comment. However, for our university, the project identification code or permission number is not available.

Point 6: Genes should be indicated in italics.

Response 6: Thanks for your comment. We have corrected the expression in main text.

Point 7: The authors should explain the relationship between obesity and the used genes (PPARγ, C/EBPα, STREBP-1c, ACC, FABP4, FAS, HSL, and PLIN) in more detail. Are these genes increased or decreased in obesity?

Response 7: Thanks for your comment. We have added more detail about those genes in main text as following:

“In obesity model, SREBP-1c, FAS, ACC and HMGCR were increased while adiponectin was decreased in liver.”

“In obesity model, PPARγ, C/EBPα, STREBP-1c, ACC, FABP4, FAS and HSL were increased while PLIN was decreased in WAT.”

Point 8: Figure 1E and F) Average molecular weights show UAOS > SAOS (page 4, line 183-184). However, the molecular weight of UAOS and SAOS determined by ESI-MS analysis exhibits the same m/z. It is a mistake?

Response 8: Thank you very much for your comment. This is not a mistake. In the ESI-MS analysis, the main peaks at 351.03 m/z and 527.04 m/z corresponded to unsaturated alginate disaccharides (DP2) and unsaturated alginate trisaccharides (DP3). This analysis used to further determine the main ingredients of UMOS and UGOS used in this study are DP2 and DP3. However, the average molecular weights of these four sugars are different because of the ratio of DP2 and DP3 are different.

Point 9: Figure 2 and 4: The mark (##) is not found.

Response 9: I am very sorry for the mistake and we have deleted the mark (##) in the legends of figure 2 and 4.

Point 10: Figure 5D: adipocyte size but not adipocytes size.

Response 10: I am very sorry for the mistake and we have changed it in figure 5D.

Point 11:References: The authors should rewrite references, for example see refs. 5, 19, 27, 34, etc.

Response 11: Thanks for your comment. We have rewritten references.

Round 2

Reviewer 1 Report

Although there is an overall amelioration in the redaction of the manuscript, the introduction and the discussion sections should be improved.

     -   In the Introduction section, the authors did not explain why AMPK is considered as a potential target for the treatment of metabolic disorders. Is AMPK activity altered by high-fat diet (HFD)-induced obesity at different tissues such as white adipose tissue? Are there any studies which have evaluated the effects of diet or any alternative method on obesity as well as on AMPK activity?

      -  In the discussion section which is too short, the results of the present study should be not only presented but also confronted to those of previous studies in order to corroborate or to explain the reasons of possible discrepancies.

The authors did not give answers to my previous questions: what is the impact of oxidative stress produced at the level of adipose tissue in the development of obesity in this animal model? Could AOSs exert beneficial impacts on obesity through their effects on adipose tissue oxidative stress?

      -    Indeed, studies have shown that obese high Glucose-fed rats showed increases in plasma levels of triglycerides as well as in superoxide anion production and NADPH oxidase activity in the adipose tissue (El Midaoui et al, Int J Mol Sci. 2017 Nov 22;18(11). pii: E2492). Moreover, antioxidant diet was found to reduce the increase in adipose tissue weight, body weight and serum triglyceride concentrations in high fat diet or high glucose fed rats suggesting that adipose tissue oxidative stress is partly implicated in the development of visceral obesity (Sour et al 2015, Metab. Cardiovasc. Dis.201525, 382–387 ; El Midaoui et al, Int J Mol Sci. 2017 Nov 22;18(11). pii: E2492).

Minor comments

In page 2, line 71, the authors stated “Increasing evidence has demonstrated that AMPK can inhibit adipogenesis and suppress the expression of SREBP-1c, PPARγ, and FAS in adipocytes [23]”. The reference 23 is not the appropriate one.

As mentioned in my last review, “UAOS (UMOS and UGOM) treatment markedly lowered serum TG and TC, and increased FFA levels” should be corrected. FFA levels were decreased not increased.

Author Response

Response to Reviewer 1 Comments

The authors investigated the effects of unsaturated alginate oligosaccharides on body weight, lipid profile, hepatic oxidative stress and AMPK signaling pathway in high-fat diet fed mouse, an animal model of obesity. This study is of interest however, some concerns should be addressed.

Although there is an overall amelioration in the redaction of the manuscript, the introduction and the discussion sections should be improved.

Point 1: In the Introduction section, the authors did not explain why AMPK is considered as a potential target for the treatment of metabolic disorders. Is AMPK activity altered by high-fat diet (HFD)-induced obesity at different tissues such as white adipose tissue? Are there any studies which have evaluated the effects of diet or any alternative method on obesity as well as on AMPK activity?

Response 1: Thank you very much for your comment. We have added more explanation in introduction section as follows: “AMP-activated protein kinase (AMPK) is a crucial cellular energy sensor. Once AMPK activated, it triggers catalytic processes to generate ATP while inhibiting anabolic processes that consume ATP in an attempt to restore cellular energy homeostasis. AMPK is considered as a potential target for the treatment of metabolic disorders [22]. It has previously been reported that anti-obesity agents activated the AMPK signaling pathway which is related to the suppression of adipogenic differentiation and lipogenesis AMPK, such as Rg1 [23], dioxinodehydroeckol [24], ethanol extracts of Aster yomena [25].”

Point 2: In the discussion section which is too short, the results of the present study should be not only presented but also confronted to those of previous studies in order to corroborate or to explain the reasons of possible discrepancies.

Response 1: Thank you very much for your kind works. We have added more explaining in discussion sections. Please see the added part as labelled in red. We wish the discussion section have improved in this version.

Point 3: The authors did not give answers to my previous questions: what is the impact of oxidative stress produced at the level of adipose tissue in the development of obesity in this animal model? Could AOSs exert beneficial impacts on obesity through their effects on adipose tissue oxidative stress?

Indeed, studies have shown that obese high Glucose-fed rats showed increases in plasma levels of triglycerides as well as in superoxide anion production and NADPH oxidase activity in the adipose tissue (El Midaoui et al, Int J Mol Sci. 2017 Nov 22;18(11). pii: E2492). Moreover, antioxidant diet was found to reduce the increase in adipose tissue weight, body weight and serum triglyceride concentrations in high fat diet or high glucose fed rats suggesting that adipose tissue oxidative stress is partly implicated in the development of visceral obesity (Sour et al 2015, Metab. Cardiovasc. Dis.2015, 25, 382–387; El Midaoui et al, Int J Mol Sci. 2017 Nov 22;18(11). pii: E2492).

Response 4: Thank you very much for your question. The opinion of reviewer is also absolutely right and we quietly agree. As reviver suggested, we will study whether the anti-obesity effect of AOSs is through anti-oxidation in adipose tissue in our further study, because we only have three days to revise manuscript this time. In addition, we are trying to study the mechanism of AOSs more deeply, such as at cellular level and intestinal flora.

Whether the anti-obesity effect of AOSs is through anti-oxidation in adipose tissue needs to be further studied. And we discussed this in discussion section. We also have added those previous studies mentioned above in discussion section. Thank you again.

Minor comments

Point 4: In page 2, line 71, the authors stated “Increasing evidence has demonstrated that AMPK can inhibit adipogenesis and suppress the expression of SREBP-1c, PPARγ, and FAS in adipocytes [23]”. The reference 23 is not the appropriate one.

Response 4: Thank you very much for your kind works. We have corrected reference #23 as new references #23-24.

Point 5: As mentioned in my last review, “UAOS (UMOS and UGOM) treatment markedly lowered serum TG and TC, and increased FFA levels” should be corrected. FFA levels were decreased not increased.

Response 5: Thank you very much for your kind works. We have corrected this wrong description in main text as following: “Moreover, compared with SAOS (SMOS and SGOS), UAOS (UMOS and UGOM) treatment markedly lowered serum TG and TC, and decreased FFA levels”.

Round 3

Reviewer 1 Report

In page 16, line 420, the authors stated “However, due to increased adipose tissue leading to obesity, whether HFD induced oxidative stress is related to obesity development and AOSs exert beneficial impacts on obesity through their effects on adipose tissue oxidative stress need to be further studied” should be replaced by “However, due to increased adipose tissue leading to obesity, whether HFD induced oxidative stress is related to obesity development in our animal model and if AOSs exert beneficial impacts on obesity through their effects on adipose tissue oxidative stress need to be further studied”.

In page 16, line 416, “which” should be deleted.

Author Response

Response to Reviewer 1 Comments

Point 1: In page 16, line 420, the authors stated “However, due to increased adipose tissue leading to obesity, whether HFD induced oxidative stress is related to obesity development and AOSs exert beneficial impacts on obesity through their effects on adipose tissue oxidative stress need to be further studied” should be replaced by “However, due to increased adipose tissue leading to obesity, whether HFD induced oxidative stress is related to obesity development in our animal model and if AOSs exert beneficial impacts on obesity through their effects on adipose tissue oxidative stress need to be further studied”.

Response 1: Thank you very much for your comment. We have modified this expression in discussion section.

Point 2: In page 16, line 416, “which” should be deleted.

Response 2: Thank you very much for your comment. We have deleted “which” in discussion section.